# The U3 and Env Proteins of Jaagsiekte Sheep Retrovirus and Enzootic Nasal Tumor Virus Both Contribute to Tissue Tropism

**DOI:** 10.3390/v11111061

**Published:** 2019-11-14

**Authors:** María C. Rosales Gerpe, Laura P. van Lieshout, Jakob M. Domm, Joelle C. Ingrao, Jodre Datu, Scott R. Walsh, Darrick L. Yu, Jondavid de Jong, Peter J. Krell, Sarah K. Wootton

**Affiliations:** 1Department of Pathobiology, Ontario Veterinary College, University of Guelph, Guelph, ON N1G 2W1, Canada; rosalesg@uoguelph.ca (M.C.R.G.); lvanlies@uoguelph.ca (L.P.v.L.); jakobdomm@gmail.com (J.M.D.); jingrao@uoguelph.ca (J.C.I.); datuj@uoguelph.ca (J.D.); darrickyu@gmail.com (D.L.Y.); dejong.jddj@gmail.com (J.d.J.); 2McMaster Immunology Research Center, Department of Pathology and Molecular Medicine, McMaster University, Hamilton, ON L8S 4L8, Canada; scott.walsh22@gmail.com; 3Department of Cellular and Molecular Biology, University of Guelph, Guelph, ON N1G 2W1, Canada; pkrell@uoguelph.ca

**Keywords:** tropism, long terminal repeat (LTR), sheep betaretroviruses, envelope glycoproteins, chimeric virus, pseudotyping, pathogenesis

## Abstract

Jaagsiekte sheep retrovirus (JSRV) and enzootic nasal tumor virus (ENTV) are small-ruminant betaretroviruses that share high nucleotide and amino acid identity, utilize the same cellular receptor, hyaluronoglucosaminidase 2 (Hyal2) for entry, and transform tissues with their envelope (Env) glycoprotein; yet, they target discrete regions of the respiratory tract—the lung and nose, respectively. This distinct tissue selectivity makes them ideal tools with which to study the pathogenesis of betaretroviruses. To uncover the genetic determinants of tropism, we constructed JSRV–ENTV chimeric viruses and produced lentivectors pseudotyped with the Env proteins from JSRV (Jenv) and ENTV (Eenv). Through the transduction and infection of lung and nasal turbinate tissue slices, we observed that Hyal2 expression levels strongly influence ENTV entry, but that the long terminal repeat (LTR) promoters of these viruses are likely responsible for tissue-specificity. Furthermore, we show evidence of ENTV Env expression in chondrocytes within ENTV-infected nasal turbinate tissue, where Hyal2 is highly expressed. Our work suggests that the unique tissue tropism of JSRV and ENTV stems from the combined effort of the envelope glycoprotein-receptor interactions and the LTR and provides new insight into the pathogenesis of ENTV.

## 1. Introduction

Jaagsiekte sheep retrovirus (JSRV) and enzootic nasal tumor virus (ENTV) are two small-ruminant betaretroviruses that infect sheep, causing ovine pulmonary adenocarcinoma (OPA) and enzootic nasal adenocarcinoma (ENA), respectively [1,2]. JSRV and ENTV share high nucleotide and amino acid identity and utilize the same cellular receptor, hyaluronoglucosaminidase-2 (Hyal2), for entry [3,4]. Their envelope (Env) glycoprotein also functions as a potent oncogene, which is rare amongst retroviruses [5,6,7]. Despite these similarities, JSRV and ENTV cause disease in different anatomical locations of the respiratory tract. JSRV targets proliferating cells of the distal lung, particularly progenitor alveolar type II pneumocytes [8]. JSRV pathogenesis depends on the availability of proliferating cells, which are more prevalent in neonatal lambs, or in sheep that have sustained damage to their lungs [8]. Additionally, JSRV is capable of infecting circulating lymphoid cells, though it replicates poorly in these cells [9,10]. In contrast, ENTV induces transformation of nasal turbinates, resulting in enzootic nasal adenocarcinoma (ENA) [2]. Unlike JSRV, ENTV does not appear to have a viremic stage [11], and currently, its pathogenesis remains incompletely elucidated [12]. The mechanism contributing to the distinct tissue specificity of JSRV and ENTV remains poorly understood. However, two areas of dissimilarity that may contribute to tropism include the long terminal repeats (LTRs), particularly the U3 promoter region, and the C-terminus of Env; the U3 differs by 37% at the nucleotide level, while the cytoplasmic tail (CT) of Env differs by 50% at the amino acid level [13]. Of note, the tissue tropisms of other betaretroviruses, including mouse mammary tumor virus and the Mason–Pfizer monkey virus, are strongly influenced by both their envelope glycoproteins and LTRs [14,15].

Currently, the consensus in the field is that ovine betaretrovirus tropism is dictated by differences in the U3 region [16], whereby ENTV or JSRV virions may be able to gain entry into a host cell yet be incapable of properly replicating due to the lack of appropriate transcription factors. Indeed, the lung and liver-specific hepatocyte nuclear factor (HNF)-3β (HNF-3β) binding site is present in only the U3 of JSRV, whereas the CCAAT enhancer binding protein (C/EBP)-α and β binding sites are conserved in the U3 of both JSRV and ENTV [16,17]. HNF-3β interacts with the JSRV LTR in alveolar type II pneumocytes, the target cell for JSRV-induced lung tumorigenesis [17,18]. However, in 2011, a study in mice found that there was no statistically significant difference in expression of the alkaline phosphatase reporter protein expressed under the control of either the JSRV or ENTV LTR when delivered to the lung using AAV vectors [18].

In addition to the LTR, the Env protein would be expected to influence tropism of the virus since it mediates interaction with the cellular receptor Hyal2 to facilitate entry [3,12]. Hyal2 belongs to a family of glycosylphosphatidylinositol–anchored cell–surface proteins responsible for degrading hyaluronic acid (HA), a major component of the extracellular matrix (ECM). While Hyal2 is widely expressed in epithelial, endothelial, and chondrocyte cells [19,20], it is enriched in the fetal ovine lung [19], which Davey et al. found to be highly susceptible to prenatal lung gene transfer with Jenv pseudotyped lentivirus (LV) [21]. Interestingly, the rat, ovine, and human Hyal2 receptors interact with Jenv and ENTV Env (Eenv) proteins, with the rat Hyal2 having a lower affinity for Eenv than for Jenv, while murine Hyal2 does not permit entry of Jenv or Eenv LV into cells [4]. Furthermore, the presence of Hyal2 at endogenous levels does not appear to be sufficient for Eenv binding, unlike Jenv; however, overexpression of Hyal2 increases Eenv-pseudotyped retroviral vector transduction [22]. Of all the viral proteins, the CT domain of Jenv and Eenv shares the lowest amino acid identity [9]. The CT of retroviruses is known to play important roles in viral assembly, replication, cellular membrane fusion, and even transformation [23]. Truncation of the CT of Jenv and Eenv results in increased transduction efficiency of pseudotyped lentivectors [24,25]; however, this enhancement is much more pronounced in the case of Eenv [24]. The differences between Jenv and Eenv are far from discrete and the mechanisms contributing to tissue tropism are possibly more complex than initially expected.

Here, we tested the hypothesis that the difference in tropism between JSRV and ENTV may be due to differences in the *env* and U3 regions of their genomes. Using chimeric viruses and lentivectors pseudotyped with Eenv and Jenv in ex vivo ovine lung and nasal turbinate tissue slice models, we found that JSRV and ENTV tissue tropism is orchestrated by the combined effort of the envelope and U3 promoter. Furthermore, we found evidence that suggests nasal turbinate chondrocytes may be a potential target cell for ENTV infection. Finally, we showed that the JSRV LTR is not as active as the ENTV LTR in ovine primary chondrocytes.

## 2. Materials and Methods

### 2.1. Cell Culture

Human Epithelial Kidney (HEK) 293, HEK 293T, ovine epithelial OA3T.s, and ovine chondrocytes (a gift from Dr. Mark B. Hurtig, Ontario Veterinary College) were cultured in high glucose Dulbecco’s modified Eagle’s medium (DMEM) (ThermoFisher, ON, Canada) supplemented with 10% Cosmic Bovine Calf Serum (BCS) (ThermoFisher), 2 mM L-Glutamine (ThermoFisher), 100 U penicillin/mL, and 100 µg/mL streptomycin (P/S) (ThermoFisher) in a humidified, 5% CO_2_ incubator at 37 °C. Murine Lung Epithelial (MLE 12) (ATCC) cells were cultured in 50:50 DMEM:DMEM/F12 (ThermoFisher), supplemented with 1g/L insulin, 0.55 g/L transferrin, and 0.67 mg/L Sodium selenite (ThermoFisher).

### 2.2. Animals

All animal experiments were conducted in accordance with the Canadian Council on Animal Care guidelines and approved by the Animal Care Committee of the University of Guelph. Six four-week old BALB/c female mice were purchased from Charles River Laboratories (QC, Canada), and housed in separate cages containing two and four mice each. The mice were acclimated to the environment for one week prior to being infected with 1 × 10^11^ vector genomes (vg) of AAV6.2FF-FTHyal2 intranasally, as described previously [26]. As negative controls, two BALB/c mice were administered vehicle alone. At three weeks post-vector administration, the mice were euthanized and their lungs harvested for generating precision-cut lung tissue slices, as described previously [27]. Specific-Pathogen Free Cornell Star neonatal lambs were purchased from the Ponsonby Research Station (Guelph, ON, Canada) and euthanized via intravenous injection of pentobarbital sodium (Euthansol, Merck Animal Health, Quebec, Canada) prior to the port-mortem procurement of lungs for precision-cut lung tissue slice generation. Nasal tissue was harvested from a still born lamb obtained from the Ponsonby Research Station.

### 2.3. DNA Constructs and PCR

The pSin-CASI-AP-WPRE plasmid was generated using the InFusion kit (Takara Clontech, Mountain View, CA, USA) according to manufacturer’s instructions. Briefly, the EF1α promoter in pSIN-EF1α was replaced by the CASI promoter, a composite promoter comprised of the cytomegalovirus immediate-early promoter (CMV), chimeric chicken-β-actin (CAG), and a ubiquitin C (UBC) enhancer region [28], using AgeI and SpeI restriction sites and the following primers: forward (FWD) 5′-cggcaattgaaccggtggagttccgcgt-3′ and reverse (REV) 5′-gccccagcatactagtctgttcgtcacccaggacct-3′. The heat-stable human placental alkaline phosphatase (AP) gene was directionally cloned using SpeI and EcoRI restriction sites and the following primers: FWD 5′-gatccttcgaactagtatgctggggccctgcatgc-3′ and REV 5′-ctcaagcttcgaattccttcagggagcagtggccgtct-3′. Finally, for increased mRNA stability and translation efficiency, the Woodchuck post-transcriptional regulatory element (WPRE) was also directionally cloned with the following primers: FWD 5′-ctccctgaaggaattcaatcaacctctggattacaaaatttgtgaaagA-3′ and REV 5′- atccttcgaactagcgctgcggggaggcggc-3′. All primers were generated using the InFusion function in the SnapGene software (GSL Biotech LLC, Chicago, IL, USA). The Q5 High Fidelity 2X Master Mix (New England Biolabs (NEB), Oshawa, Ontario, Canada) was employed to generate the CASI, hPLAP, and WPRE inserts, according to manufacturer’s specifications, with a few exceptions. For PCR amplification of the CASI and WPRE fragments, betaine and dimethyl sulfoxide were added to reduce potential secondary structures and the extension time was increased to 1 min. The PCR program used was: 1 cycle at 95 °C for 30 s, 30 cycles of 95 °C for 5 s, 60 °C for 15 s, and 72 °C for 20 s, and a final cycle of 2 min at 72 °C. Following PCR, the products were subjected to a 1 h DpnI digestion at 37 °C to remove template plasmid DNA. The NucleoSpin Gel and PCR Clean-up Extract II kit was used to purify the DNA after PCR amplification and gel electrophoresis, as indicated by the manufacturer (Fisher Scientific).

The chimera molecular clones were generated via overlap extension PCR (OE PCR) following a previously published protocol [29]. Primers were first used to create amplicons termed ‘inserts’ for the chimeras using Q5 HF DNA polymerase (NEB, Ontario, Canada), that were then inserted via OE PCR using the non-strand displacement Phusion polymerase (NEB), both according to manufacturer specifications. The primers used to generate inserts for chimeras are listed in Appendix A.

Two different programs were used to generate inserts 1, 2, 4, and 5 (1 cycle of 30 s at 98 °C, 30 cycles of 5 s at 98 °C, 30 s at 62 °C, and 60 s at 72 °C, and 1 cycle of 2 min at 72 °C) and inserts 3 and 6 (1 cycle of 30 s at 98 °C, 30 cycles of 5 s at 98 °C, 30 s at 65 °C, and 60 s at 72 °C, and 1 cycle of 2 min at 72 °C). Two other programs were used to perform overlap extension PCR for Chm1-ERU5, Chm2-Egag, Chm4-EU3, and Chm5-ELTR (1 cycle of 30 s at 98 °C, 20 cycles of 5 s at 98 °C, 30 s at 60 °C, and 18 min at 72 °C, and 1 cycle of 10 min at 72 °C), and for Chm3 and 6 (1 cycle of 30 s at 98 °C, 18 cycles of 5 s at 98 °C, 30 s at 60 °C, and 36 min at 72 °C, and 1 cycle of 10 min at 72 °C) with the addition of 5 µm betaine and 4% DMSO. Finally, site-directed mutagenesis was performed on Chm3-Eenv to remove the original splice acceptor site for Jenv in the JSRV backbone using Phusion HF DNA polymerase with the following program—1 cycle of 30 s at 98 °C, 25 cycles of 10 s at 98 °C, 30 s at 70 °C, and 7 min at 72 °C, and 1 cycle of 2 min at 72 °C—and the following primers: FWD 5′-gtttgtgtttttccacaaaatgccgaagcgccg-3′, and REV: 5′-cggcgcttcggcattttgtggaaaaacacaaac-3′.

The pACAG-PPT-FT-Hyal2 vector was directionally cloned into restriction sites SphI and XhoI (NEB) in the pACAG-MCS vector, as previously described [30]. The Hyal2 CDS was amplified from the pL-Hyal2-SN plasmid [4] using Q5 High Fidelity 2X Master Mix (NEB) and the following primers: FWD—5′-tctcgcatgcatgtctgcacttctgatcc-3′ and REV—5′-cacactcgagctacaaggtccaggtaaagg-3′, to include a preprotrypsin (PPT) signal before the FLAG tag in the N-terminus of hHyal2. The following PCR program was used to generate the PCR product: one cycle at 98 °C for 30 s, followed by 35 cycles of 98 °C for 5 s, 62 °C for 30 s, and 72 °C for 60 s, and one final cycle of 72 °C for 2 min.

### 2.4. Virus Production

Lentivectors and virus particles were produced in HEK 293T and HEK 293 cells, respectively. Two million cells were seeded per 10-cm dish for a total of fifteen 10-cm dishes. Twenty four hours post-seeding, cells were transfected using the following per dish: 5 µg of genome plasmid (pSinGFP or pSinCASI-hPLAP-WPRE), 3.25 µg of helper plasmid (psPAX2), and 1.75 µg of envelope plasmid (EBOV, VSVg, GP64, Eenv, and Jenv) to generate LVs or 10 µg of plasmids: pCMVJS21 for JSRV [1], pCMVENTV-1NA4 for ENTV [31], pCMVJS21-ENTVRU5 for Chm1-ERU5, pCMVJS21-ENTVgag for Chm2-Egag, pCMVJS21-ENTVenv for Chm3-Eenv, pCMVJS21-ENTVU3 for Chm4-EU3, and pCMVJS21-ENTVFullLTR for Chm5-ELTR. A total of 67.5 µL of 1 mg/mL polyethylenimine transfection polymer, and 450 µL of DMEM were also used per dish. Two hours post-transfection, the media was changed with DMEM supplemented with 2 mM L-glutamine, and P/S (basal DMEM). Every 24 h, for a total of 72 h, the supernatant was collected into 50 mL conical tubes and replenished with fresh basal DMEM. Once collected, the supernatant was spun down in a swinging-bucket centrifuge at 500× *g* for 5 min to pellet cellular debris. The clarified supernatant was passed through a 0.45 μm polyethersulfone syringe filter (VWR, Mississauga, Ontario, Canada) and stored at 4 °C until all three LV collections were procured. Virus was concentrated with polyethylene glycol (PEG-8000) in a 1:2 PEG:supernatant ratio, and kept rocking at 4 °C overnight before spinning the next day in a swinging-bucket centrifuge at 4000× *g* for 15 min. Pellets were re-suspended in Tris-NaCl-EDTA (TNE) buffer, pH 7.4 (100 mM NaCl, 10 mM Tris-HCl, and 1 mM EDTA).

### 2.5. Virus Quantification, Transduction, and Infection

LV titers were measured using the Lenti-X™ p24 Rapid Titer Kit (Takara Bio USA, Inc.). Reverse transcriptase (RT) activity was measured using the EnzChek^®^ Reverse Transcriptase Assay (ThermoFisher Scientific). Post-production of particles, HEK 293T (for LVs) or HEK 293 (for JSRV, ENTV and the chimeras) cells were lysed, or their supernatant was harvested for ultracentrifugation prior to lysing with RIPA buffer and processed for western blot or EM. Protein expression was visualized via WB and vector particles were imaged using EM, as described previously [31]. JSRV, ENTV and the chimeras were titered via RT assay and p27 ELISA. Tissue slices were infected with 1 × 10^6^ infectious units (IFU), as previously described [27].

### 2.6. Western Blotting

To detect the Env of JSRV and ENTV, 1:10 mouse hybridoma supernatant against the SU of the JSRV and ENTV Env (DM12 [32]) and 1:3000 goat anti-mouse secondary antibody were used. For a loading control, a primary rabbit anti-β-actin (Cell Signaling 4970S) antibody was used at 1:1000 with a secondary goat anti-rabbit HRP-conjugated antibody at 1:5000. To detect the FLAG tagged hHyal2, Rabbit α-FLAG (Abcam, catalogue number Ab21536) was used at 1:1000 with the goat anti-Rabbit HRP-conjugated antibody at 1:5000.

### 2.7. Immunization Protocol to Generate Murine Anti-p27 Serum

Four six-week-old BALB/c mice were immunized using a protocol designed based on previous studies [33,34]. Each mouse was immunized with 25 µg of recombinant p27 protein [31] and AdjuPhos in a 1:3 v/v antigen: adjuvant ratio, with the addition of 10 µg of QuilA. A total of 50 µL was administered to each flank, for a total of 200 µL of antigen-adjuvant solution per mouse. Animals were boosted three times with 10 µg of antigen and the same adjuvant proportions, as described above. Appendix A outlines the immunization schedule.

### 2.8. ENTV and JSRV p27 ELISA

An immunolon 2HB plate (Fisher, ON, Canada) was coated with 200 µL of 1:100 hyperimmune serum from mice challenged with recombinant p27 capsid protein diluted in fresh coating buffer (0.015 M Sodium carbonate, 0.036 M Sodium bicarbonate, pH 9.6) and left to incubate overnight at 4 °C. The wells were washed with wash buffer (0.5% v/v Tween-20 in 1X PBS) before blocking with 200 µL of 3% bovine serum albumin (BSA) in 1× PBS at 37 °C for 1 h and 30 min while rocking in a hybridization incubator. The plate was then washed three times with 200 µL of wash buffer before 100 µL of samples and standards were added and incubated for 1 h and 30 min at 37 °C while rocking. The plate was then washed as described before, and 100 µL of 1:50 rabbit serum anti-ECa [31] was added per well and incubated for 1 h and 30 min at 37 °C while rocking. The plate was washed again as mentioned. Anti-rabbit IgG HRP (Cell Signaling, NEB, Cambridge, MA, USA) (100 µL per well) was then added at 1:1000 and for 1 h at 37 °C with rocking. The plate was washed for a final time as detailed. Post-washing, 150 µL of 2,2’-Azinobis (3-ethylbenzothiazoline-6-sulfonic acid)-diammonium salt (ABTS) substrate was added per well and the whole affair was incubated for 30 min at room temperature before adding 100 µL of 1% *w*/*v* SDS to stop the reaction. The plate was read at an absorbance of 405 nm.

Virus samples were prepared by lysing with soft lysis buffer (1% *v*/*v* Triton-X, 25 mM Tris-HCl pH 7.5, 150 mM NaCl, and 5 mM EDTA) at 1:10 lysis buffer:sample. Lysed samples were then diluted to 1 in 8 with wash buffer. A standard curve was set up using ECa at the following concentrations: 10 µg/mL, 5 µg/mL, 2.5 µg/mL, 1.25 µg/mL, 0.625 µg/mL, 0.313 µg/mL, and 0.157 µg/mL. Concentrations of JSRV, ENTV, and the chimeras, were measured by calculating infectious units using the following two assumptions: 1 ng of p27 represents 1.25 × 10^7^ retroviral particles, and one infectious unit (IFU) represents 1000 retroviral particles, based on previous research with LV particles [35].

### 2.9. Immunohistochemistry (IHC)

Lung lobes from PBS-treated control and AAV6.2FF-FT-Hyal2 infected (1 × 10^11^ vector genomes (vg)) BALB/c mice were harvested three weeks post-vector administration. Hyal2 expression was detected by immunohistochemical staining for FLAG using rabbit α-FLAG (Abcam, catalogue number Ab21536) at 1:100 in 3% bovine serum albumin in 1X PBS-Tween-containing SignalStain^®^ Boost IHC Detection Reagent (HRP Rabbit) (Cell Signaling, NEB) as described [36]. To detect the Env of JSRV and ENTV, 1:50 mouse DM12 antibody and 1:100 goat anti-mouse secondary antibody were used, as previously described [32]. SIGMAFAST^TM^ 3,3′-Diaminobenzidine DAB (Sigma-Aldrich, Saint Louis, MO, USA) was used for detection following the manufacturer’s instructions. The slides were counterstained with hematoxylin and mounted with Richard–Allan ScientificTM Cytoseal XYL (Thermo Fisher Scientific), before being imaged with a Clinical Olympus BX45 microscope (Olympus, Tokyo, Japan). Pathologists Dr. Joelle Ingrao, Josepha DeLay, and Maria Spinato of the Ontario Veterinary College identified cell types based on hematoxylin and eosin (H&E) staining. AP staining of LV transduced lung tissue slices from AAV6.2FF-FT-Hyal2 infected mice was performed as described previously [30].

### 2.10. Statistical Analyses

Statistical analyses were performed using GraphPad Prism 7 software (GraphPad, La Jolla, CA, USA) and standard errors were calculated for all figures (error bars).

## 3. Results

### 3.1. Construction of JSRV and ENTV Hybrid Viruses

To ascertain the region of the viral genome responsible for tissue specificity, six JSRV–ENTV hybrid viruses or chimeras (chm) were engineered within the backbone of the JSRV molecular clone (Figure 1A) using overlap extension PCR: Chm1-ERU5 (ENTV R+U5), Chm2-Egag (ENTV gag), Chm3-Eenv (ENTV Env), Chm4-EU3 (ENTV U3), and Chm5-ELTR (ENTV full LTR) (Figure 1B). Western blot analysis of viral and cellular lysates from cells transfected with the chimeric molecular clones demonstrated production of mature virus particles, as evidenced by the presence of Env SU and fully processed capsid protein (Figure 1B). Western blots on cell lysates and ultra-centrifuged, virus-containing supernatants (viral lysates) were performed using an antibody targeting the surface (SU) domain of Env (anti-SU antibody) to detect Env [32] and an anti-p27 antibody [31] to detect the JSRV and ENTV capsids. Western blotting with anti-Env revealed the presence of a roughly 63 kDa protein, the predicted size of the SU domain of Jenv and Eenv. Note that this antibody does not readily detect unprocessed Env or the TM domain. We observed a slight difference in molecular weight between Eenv and Jenv in the western blots of lysates from cells transfected with JSRV, chimeras encoding Jenv (Chm1-ERU3, Chm2-Egag, Chm4-EU3, and Chm5-ELTR) and ENTV, and a chimera encoding Eenv (Chm3-Eenv); and in the viral lysates from these samples. Additionally, a protein band migrating at roughly 27 kDa was detected in all transfected cell and viral lysates after anti-p27 immunoblotting, although only weakly for Chm1-ERU5 (viral lysate) and Chm2-Egag (cell lysate). Electron microscopy of the supernatants from HEK 293 cells transfected with the various chimeric molecular clones showed evidence of retrovirus-like particles (Appendix A), in some cases with eccentric cores and an average size of 100–150 nm, consistent with other betaretroviruses [37].

### 3.2. Jenv-Encoding but Not Eenv-Encoding Chimeras Are Able to Infect Ovine Lung Tissue Slices

Having determined that the JSRV–ENTV chimeric viruses were viable, we sought to investigate which regions of the genome were important for mediating the infection of ovine lung slices. Viruses were titered using a reverse transcriptase activity assay in combination with an in house p27 ELISA. As there were no commercial ELISAs available to quantify JSRV or ENTV, we developed a sandwich ELISA targeting the p27 capsid protein (Appendix A), similar to the HIV p24 ELISA. Using anti-p27 sera derived from mice immunized with recombinant p27 capsid protein along with previously-produced rabbit anti-p27 hyperimmune sera [31], functional or processed virions were quantified. The amounts of JSRV, ENTV, and chimeric viruses produced from HEK 293T cells ranged between 1 × 10^3^ and 8 × 10^3^ pg/mL of p27, which is approximately 1 × 10^3^–7 × 10^7^ infectious units per mL (IFU/mL) [38] (Appendix A). Ovine lung tissue slices were infected with 1 × 10^6^ IFU of JSRV, ENTV and the six chimeric viruses described in Figure 1. Three weeks post-infection, we detected infection of ovine lung tissue slices with JSRV, ENTV, and the chimeras by immunohistochemical staining for Env. Positive immunostaining (brown) representative of Env expression was observed in ovine lung tissue slices infected with JSRV and chimeras expressing the envelope of JSRV (Chm1-ERU5, Chm2-Egag, Chm4-EU3, and Chm5-ELTR) but not with ENTV or chimeras encoding Eenv (Chm3-Eenv) (Figure 2). No positive immunostaining was observed in uninfected lung tissue slices. Taken together, these results suggest that the JSRV Env protein is capable of mediating viral entry into ovine lung tissue slices, whereas the ENTV Env protein is not.

### 3.3. ENTV Infects Ovine Nasal Turbinate Tissue Slices with Much Greater Efficiency Than JSRV

Based on the results we observed with the ovine lung tissue slices, we decided to investigate the infection of fetal ovine nasal turbinate tissue slices. In order to generate nasal tissue slices, fetal lamb tissue was required, as only this nasal tissue was amenable to slicing with our vibratome. We were able to obtain a small number of viable nasal tissue slices from an infected, aborted fetus, and these were subsequently infected with with 1 × 10^6^ IFU of JSRV, ENTV, and chimeras 2, 3, and 4. Chm3-Eenv and Chm4-EU3 were used to evaluate the roles of Eenv and ENTV U3 in the nasal tract, given that the Env and U3 share the lowest nucleotide and amino acid identities, respectively [13]. Chm2-Egag, encoding ENTV gag, which shares >90% amino acid identity with JSRV gag [13], was used as a negative control for nasal turbinate tissue infection. At three weeks post-infection, the supernatant was collected and RT activity measured (Figure 3A). Supernatants of JSRV and Chm2-Egag-infected ovine nasal turbinate slices had statistically significantly lower RT activity than the supernatant of ovine nasal turbinate slices infected with ENTV, Chm3-Eenv, and Chm4-EU3 (Figure 3A). Moreover, IHC staining for the SU of Env in the fixed nasal slices demonstrated strong expression of Env in ovine nasal turbinate slices infected with ENTV and Chm3-Eenv, while no obvious staining was observed in ovine nasal slices infected with JSRV (Figure 3B). Interestingly, the pattern of staining suggested that cartilage tissue was infected by ENTV and Chm3-Eenv, as evidenced by the brown staining in chondrocytes. Finally, using IHC to detect the Env SU, we were able to observe some positive chondrocyte staining in a clinical sample of naturally occurring enzootic nasal adenocarcinoma (ENA) (Figure 3C). Positive staining indicative of the presence of the Env SU was observed in the extracellular matrix of chondrocytes, which is known to be enriched with HA, and also serves as a reservoir for HA degradation enzymes such as Hyal2, and the nuclei of the chondrocytes [19,20,39,40]. Hematoxylin and eosin staining of the cartilage suggested histological changes consistent with degeneration, including calcification in the ENA sample (Figure 3C). Pathologists, Drs Joelle Ingrao, Josepha DeLay, and Maria Spinato identified chondrocytes in this tissue and confirmed their positive staining for Env SU.

### 3.4. Jenv Pseudotyped LV Transduces Ovine Lung Tissue Slices with Greater Efficiency Than Eenv Pseudotyped LV

To investigate the role of Env in mediating transduction of ovine lung tissue in the absence of any other viral proteins, we evaluated the transduction efficiencies of Jenv and Eenv-pseudotyped LVs in ovine lung tissue slices. Lung slices were infected with 1 × 10^6^ IFU of each LV in the presence of polybrene and imaged 48 h later (Figure 4A). VSVg and GP64-pseudotyped LVs were included as controls, whereas EBOV LV was excluded from this experiment because we had previously found that LVs pseudotyped with the EBOV glycoprotein do not appreciably transduce ovine lung tissue [41]. No green punctate foci, representing cells transduced with GFP-expressing LV (GFP channels), were present in the ovine lung tissue slices transduced by Eenv-pseudotyped LV (Figure 4A). In contrast, VSVg and Jenv-pseudotyped LVs were capable of transducing ovine lung tissue slices, as evidenced by the presence of green puncta (Figure 4A). Quantification of GFP positive foci revealed statistically significantly greater transduction of ovine lung tissue slices with Jenv LV compared to Eenv LV (Figure 4B).

### 3.5. Eenv-Pseudotyped Lentivector Entry Is Enhanced in Cells Overexpressing hHyal2

Given the results of the ex vivo ovine lung slice experiment, we were interested to determine whether overexpression of Hyal2 could enhance Eenv-LV entry. To do so, we employed HEK 293T cells, since they express endogenous levels of Hyal2 that permit Jenv-LV entry, but are poorly transduced by Eenv-LV [42] and a murine alveolar type II cell line (MLE 12) which does not permit entry to either Jenv or Eenv LV. Duplicate wells of HEK 293T and MLE 12 cells (*n* = 3) were co-transfected with a plasmid expressing Flag tagged human Hyal2 (FT hHyal2) and another expressing firefly luciferase, as a control for transfection efficiency. Forty-eight hours post-transfection, one set of mock or FT hHyal2 transfected cells were transduced with 1 × 10^6^ IFU of Jenv-LV, Eenv-LV, and EBOV-LV, and the second set was used for immunoblot analysis and quantification of firefly luciferase activity. Western blot results showed expression of FT hHyal2 at varying levels (Figure 5A). To compare the transduction efficiency across samples, the transfection efficiency was normalized according to luciferase expression and the number of GFP positive cells relative to luminescence was plotted. After this adjustment, the data showed that Eenv LV transduced hHyal2-expressing HEK 293T cells just as efficiently as Jenv LV and that expression of hHyal2 murine lung epithelial cells can permit transduction of both Jenv and Eenv pseudotyped LVs (Figure 5B).

Given that a murine lung epithelial cell line overexpressing hHyal2 became permissive to transduction by Jenv and Eenv LVs, we wanted to investigate whether the same would be true in an ex vivo lung slice murine model. Briefly, 1 × 10^11^ vg of AAV6.2FF-FT-hHyal2 was administered to 6-week old Balb/c mice intranasally. After three weeks, the lungs were harvested and used to generate murine lung slices stably expressing FT hHyal2. Immunohistochemical staining confirmed robust and widespread expression of FT hHyal2 in the lung parenchyma (Figure 5D). Two days after transducing the FT-hHyal2-expressing murine lung tissue slices with 1 × 10^6^ IFU of Eenv, Jenv, and EBOV-pseudotyped LV expressing the heat-stable human placental *alkaline phosphatase* (*AP*) reporter gene, lung slices were heat inactivated to ablate endogenous AP expression and stained for heat-stable AP expression. Numerous clusters of dark purple cells were detected in the transduced tissues, but not in the control mouse lung slice that did not receive AAV6.2FF-FT-hHyal2. Importantly, AP staining was observed in both the Jenv LV and Eenv LV-transduced tissues, demonstrating that over-expression of Hyal2 is sufficient to permit transduction of these pseudoytped LVs in murine-derived ex vivo lung slices (Figure 5E).

### 3.6. The ENTV LTR Is Significantly more Active in Ovine Primary Chondrocytes Than the JSRV LTR

Given the observed tropism for chondrocytes in ENTV infected nasal turbinate slices, which are known to be enriched in Hyal2 [16], next, we investigated whether Eenv pseudotyped LV expressing GFP might transduce primary ovine chondrocytes with greater efficiency than Jenv LV. Surprisingly, we found that Eenv and Jenv LV transductions of primary chondrocytes were not significantly different (Figure 6A). Based on this result, we turned our attention to the LTRs of these viruses, which have been postulated to dictate tissue specificity [11]. Duplicate wells of primary ovine chondrocytes (performed in triplicate) were co-transfected with plasmids encoding the AP reporter gene under the control of the CASI promoter [31], the JSRV LTR (JwtAP), or the ENTV LTR (EwtAP) in combination with a GFP expression vector as a transfection control. Previously, we identified an enhancer sequence, designated JE in the case of JSRV or EE in the case of ENTV, comprised of the 75 nucleotides upstream of the 3’ LTR within the *env* gene [30] (see Figure 1A), in a region that has been shown to be dispensable for transformation [43], but possesses enhancer functions in other retroviruses [44,45]. Since these enhancers were shown to increase tissue specific gene expression from their respective LTRs, we included JSRV LTR + JE enhancer (JEnhancerAP) and ENTV LTR + EE enhancer (EEnhancerAP) constructs in this assay. GFP expression was measured via flow cytometry and promoter activity was measured by counting the number of AP positive cells using the brightfield channel. We then generated a ratio of these two values to represent normalized AP expression (Figure 6B). The JSRV LTR and JSRV LTR + enhancer constructs were not significantly different from the CASI-AP control, whereas the ENTV LTR and the ENTV LTR + EE enhancer promoted significantly greater reporter gene expression in primary ovine chondrocytes than the JSRV LTR and JSRV LTR + enhancer constructs.

## 4. Discussion

To understand which genomic region is responsible for the difference in tissue specificity between JSRV and ENTV, six chimeric viruses were generated using JSRV as a backbone and swapping JSRV regions for the equivalent sequences from ENTV-1. The chimeras were designed to contain sequences responsible for important aspects of the betaretroviral replication cycle [46], such as replication and packaging (Chm1-ERU5–ENTV R+U5), packaging and structural proteins (Chm2-Egag–ENTV gag), cell entry (Chm3-Eenv–ENTV env), transcription (Chm4-EU3–ENTV U3), and finally replication, packaging, and transcription (Chm5-ELTR–ENTV full LTR). We also utilized LVs pseudotyped with Jenv and Eenv to better discern the role of the Env glycoprotein in determining tropism. Furthermore, we were able to utilize a novel p27 ELISA to measure JSRV and ENTV virus concentrations.

Ex vivo tissue slices are remarkably useful models that permit the study of certain aspects of the pathogeneses of infectious agents, including tropism [47,48,49]. Our ex vivo ovine lung and nasal turbinate tissue slices permitted us to overcome the inabilities of JSRV and ENTV to infect cellular monolayers. Furthermore, we utilized a novel murine model previously developed in our lab that overexpresses human Hyal2 in the lung to investigate JSRV and ENTV tissue specificity. This is also the first study to show Eenv-pseudotyped LV transduction of ovine chondrocytes in vitro, and ENTV infection of ovine nasal turbinate chondrocytes ex vivo and in vivo. Previous studies have focused on the ovine lung or have employed ovine lung epithelial cells [21,49,50,51]. Our data demonstrate that both the Env and U3 are required for JSRV and ENTV tissue selectivity, and that overexpression of Hyal2 might be crucial for ENTV infection, which may initiate within nasal turbinate chondrocytes, where Hyal2 is enriched [19,39].

Studies with other retroviruses have shown that LTR-driven transcription and the viral envelope both affect viral tropism [52,53,54,55,56,57]. While the Env protein plays a crucial role in recognizing and binding to specific receptors and mediating the fusion, LTRs may confer tropism to certain cell types by allowing more efficient replication of the viral genome in these cells. In the case of the closely related mouse mammary tumor virus (MMTV), expression of transferrin receptor 1 (TfR1), the MMTV entry receptor, contributes in part to the in vivo tissue-specific tropism of this virus, since activated cells of the immune system and dividing mammary epithelial cells express high levels of this protein [58,59,60]. However, like JSRV and ENTV, cell-type restriction in vivo is also due to events that occur post-entry. For example, the enhancer elements in the LTR function primarily in mammary epithelia and lymphoid cells, and thus, MMTV is not transcribed in many tissues [61] and particular rearrangements in the U3 region of the LTR can dramatically change the target tissue of transformation from mammary epithelial to lymphoid cells [62]. As with the findings reported for MMTV, both the Env and LTR regions contribute to the distinctive tissue tropisms of JSRV and ENTV.

Eenv-pseudotyped LVs had lower transduction efficiencies than Jenv-pseudotyped LVs in HEK 293T cells and MLEs transfected with FT hHyal2. Other studies have previously demonstrated that overexpression of hHyal2 in different cell lines enhances transduction by Eenv-pseudotyped LV [4,12,22], with the exception of rat cells [22]. However, although binding efficiency between Hyal2 and the Eenv surface glycoprotein was measured in the study by van Hoeven and Miller in 2005, the relative abundance or expression of Hyal2 was not measured or compared between cell lines due to lack of appropriate antibodies [22]. In addition, our data showed that when Hyal2 was overexpressed in transfected HEK 293T cells, Eenv-pseudotyped lentivectors were able to transduce these cells to the same extent as Jenv-pseudotyped LV. Furthermore, there was no statistically significant difference in transduction efficiency between Eenv and Jenv pseudotyped LVs’ transductions of ovine chondrocytes.

Interestingly, the level of Env SU protein detected in chm3 (which contains the Eenv protein) was much lower than those of chm1, 2, and 4 (which contain the Jenv protein), but levels of p27 in these chimeras were similar (Figure 1B). One possibility is that the Eenv protein is less efficiently incorporated into viral particles than Jenv, and thus, Eenv-containing viruses require a higher amount of Hyal2 receptor protein to promote attachment and entry.

Studies have shown that JSRV-like viruses (enJSRVs) invaded the germline of sheep at least 5–7 million years ago, leading to the presence of more than 27 endogenous betaretroviruses within the ovine genome [63,64]. While most of the research on endogenous ovine betaretroviruses has focused on JSRV and the female ovine reproductive tract [65], it would be interesting to determine whether ram testes harbor endogenous ENTV-like sequences since HA metabolism is important in the testes and Hyal2 might be highly expressed in this tissue [66]. Indeed, Eenv-pseudotyped LVs efficiently transduced OA3T.s cells, a continuous cell line derived from ovine testes (Appendix A).

The presence of JSRV-like and ENTV-like endogenous betaretroviral sequences in the sheep genome that share high nucleotide and amino acid identities with exogenous JSRV and ENTV [59] has tolerized the sheep immune system against these exogenous viruses [67,68]. Due to the lack of seroconversion upon infection with exogenous JSRV or ENTV, efforts to generate an ELISA that detects the presence of JSRV or ENTV virus particles in infected sheep continue [69,70]. In 2004, Marozsan et al. showed that RT activity could correlate with ELISA values [38]. We also found that our p27 ELISA values were similar to our RT activity values. In addition, our estimated infectious units based on our p27 ELISA values are in line with infectious unit values of other retroviruses produced under similar conditions [71].

We observed uniform overexpression of human Hyal2 in murine lung tissue after intranasal delivery of AAV-FT-hHyal2, which permitted similar transduction levels of LVs pseudotyped by Eenv and Jenv. The use of LVs allowed us to conclude that high levels of Hyal2 are crucial for the cellular entry of Eenv-containing viruses. Moreover, newborn ovine lung tissue slices were neither transduced by Eenv-pseudotyped LV nor infected by ENTV or Eenv-encoding chimeras. Interestingly, previous studies have shown that fetal lungs are enriched with Hyal2 [72], but that this declines rapidly after birth [21]. However, this level might not meet the threshold level of Hyal2 needed for ENTV infection. Indeed, our results would suggest that in the case of ENTV, an endogenous threshold of Hyal2 seems to act as the primary block to infection in the lung. It would be interesting to see whether ENTV LTRs also play a significant role in restricting ENTV spread in the ovine lung by administering an AAV vector expressing Hyal2 to a sheep lung and then infecting it with Eenv-pseudotyped LV, Chm3-Eenv or Chm5-EU3.

We observed higher RT activity in the supernatant of nasal turbinate tissue slices infected with ENTV, Chm3-Eenv, and Chm4-EU3 than in those infected with JSRV and Chm2-Egag. We also detected robust staining in nasal tissue infected with ENTV and Chm3-Eenv, compared to the faint staining for Env in the cartilage of JSRV-infected nasal turbinate tissue slices. Moreover, our data showed Env-stained chondrocytes in ENA tissue but not in normal tissue. Our results suggest that ovine nasal tissue is permissive to Jenv-mediated entry of JSRV, but that JSRV does not seem to replicate in this tissue as efficiently as ENTV or Chm3-Eenv. It is important to note that although Chm3-Eenv only encodes the ENTV Eenv, the cDNA of Eenv extends into the U3. It is possible that important ovine nasal tissue transcription factors could bind to this region permitting ENTV and restricting JSRV viral transcription. Notably, our lab previously demonstrated that the C-terminus of Eenv and Jenv contain enhancer elements [30]. These enhancer elements are located at the 3′ end of the *env* gene, ~75-bp upstream of the 3′LTR, in a region which has been shown to be dispensable for transformation [43], but possesses enhancer functions in other retroviruses [44,45]. However, there was no significant difference observed in AP expression when under the control of the JSRV LTR or the JSRV LTR plus enhancer in primary ovine chondrocyte cells. In contrast, the ENTV LTR induced significantly higher AP expression than the JSRV LTR. The expression of AP under the control of the ENTV LTR plus enhancer was also significantly higher than under the control of the ENTV LTR alone.

Previous studies have painted a complex picture where JSRV and ENTV tropism may rely on receptor utilization [22], specific pH conditions [24,25,73], and JSRV and ENTV promoter regions [16,30], some of which we have shown in this paper. Particularly, pH seems to be an interesting aspect that connects some of the aforementioned factors and may be important for Env-related tissue selectivity. Fusogenicity to intracellular membranes by Eenv relies on a much lower pH (pH 4.5) than Jenv (pH 6.0) [25,73]. Interestingly, compared to the lung, the pH of the nose is lower [74,75]. HA metabolism also relies on low pH levels (pH < 5) [76] and Hyal2 HA-degradation activity is turned on at a low pH (pH 6.0) [77,78]. Moreover, low pH promotes a tumorigenic environment by activating HA-degradation enzymes that foster tumor cell invasion [79].

Studies have shown that most of the cartilage is destroyed in nasal turbinates burdened by ENA, but this has been explained by the invasion of ENA into neighboring tissue [2]. Perhaps, expression of Eenv in ENTV-infected chondrocytes could enable a low pH environment in vivo, precipitating tumorigenesis. Given that chondrocytes exhibit a low turnover rate, future in vivo studies using JSRV, ENTV, Chm3-Eenv, and Chm4-EU3 in sheep will be crucial to conclude whether ENTV begins replicating in the nasal tract via epithelial cells or chondrocyte-precursor cells, and whether cartilage or epithelial tissue is initially transformed.

## Figures and Tables

**Figure 1 viruses-11-01061-f001:**
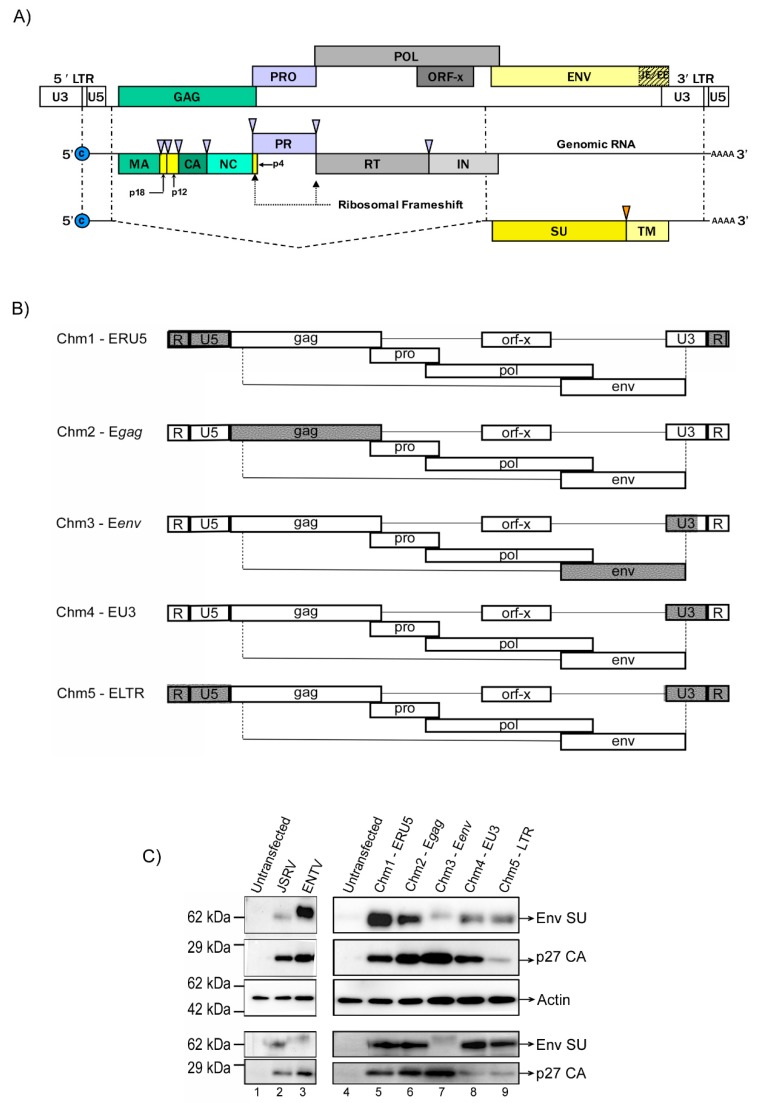
Production of Jaagsiekte sheep retrovirus (JSRV) and enzootic nasal tumor virus (ENTV) chimeric viruses (JSRV–ENTV). (**A**) The ovine betaretrovirus provirus is shown annotated with all ORFs, including Gag, Pol, Pro, and Env, and flanked by the long terminal repeats (LTRs). An additional open reading frame X (Orf-X) is present within the Pol region with unknown function. Genomic mRNA and a singly spliced transcript are shown with the recognized protein products of each transcript. Loci of ribosomal frameshift to produce fusion Gag–Pro and Gag–Pro–Pol polyproteins are indicated by large black arrowheads. Purple arrowheads indicate viral protease cleavage sites. The orange arrow indicates the host protease’s cleavage site. The JSRV/ENTV enhancer sequence (hashed box labelled JE and EE) is comprised of a 75 bp sequence located immediately upstream of the U3. (**B**) Schematic representation of JSRV–ENTV hybrid viruses within the JSRV backbone. Regions from ENTV (in grey) were swapped for complementary regions in JSRV (white). (**C**) Representative western blot of cell and viral lysates of JSRV, ENTV, and chimeric viruses produced in HEK 293 cells showing expression of the envelope (Env SU) and capsid (p27) proteins.

**Figure 2 viruses-11-01061-f002:**
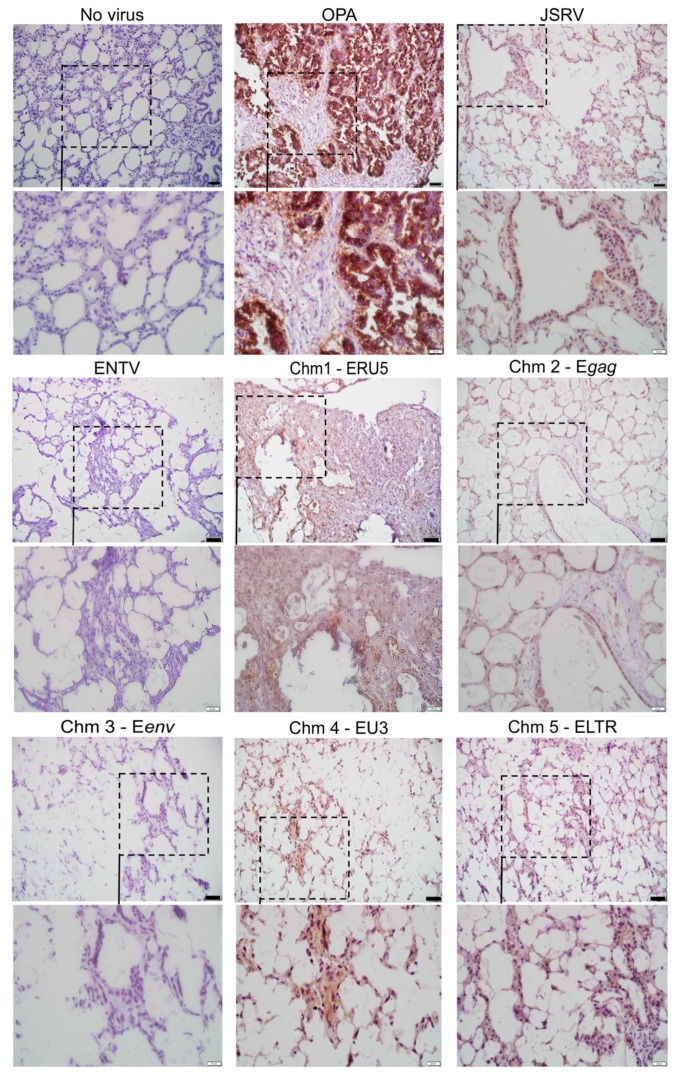
Chimeric viruses possessing the JSRV Env are capable of infecting ovine lung tissue slices. Paraffin-embedded ovine lung tissue from a field sample of ovine pulmonary adenocarcinoma (OPA; positive control) or paraffin-embedded ovine lung tissue slices infected with 1 × 10^6^ IFU of JSRV, ENTV, and the chimeras, were subjected to immunohistochemical staining for Env protein expression three weeks post-infection. Representative images show positive immunostaining (brown) for the Env protein of JSRV and ENTV in the positive OPA control, and the JSRV, Chm1-ERU5, Chm2-Egag, Chm4-EU3, and Chm5-LTR infected lung slices. Dotted boxes in the image represent the magnified field. Black boxes represent 50 µm and white boxes represent 20 µm.

**Figure 3 viruses-11-01061-f003:**
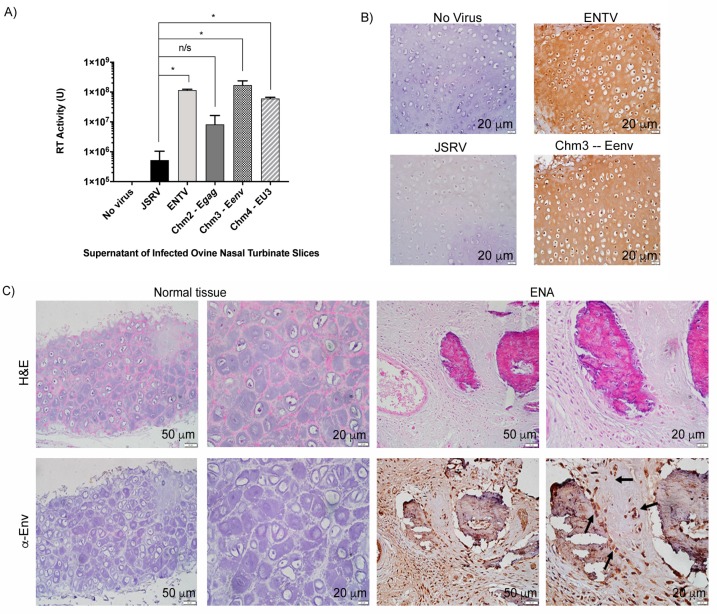
ENTV infects ovine nasal turbinate slices with greater efficiency than JSRV. (**A**) Reverse transcriptase (RT) activity of supernatant obtained from ovine nasal turbinate slices infected with 1 × 10^6^ IFU of JSRV, ENTV, and chimeras 2–4 three weeks post-infection. (**B**) IHC staining of ovine nasal turbinate slices for the Env of JSRV and ENTV shows positive immunostaining for the Env protein in the extracellular matrixes, cytoplasms, and nuclei of chondrocytes (brown) in ENTV and Chm3-Eenv-infected tissue slices, but not in JSRV-infected tissue slices. (**C**) IHC of nasal turbinate from a normal sheep and a sheep with enzootic nasal adenocarcinoma (ENA) showing positive immunostaining for Eenv in the extracellular matrix, cytoplasm, and nuclei of chondrocytes, and in the cytoplasm and nuclei of epithelial cells, as indicated by the black arrows. Unpaired student’s *t*-tests were performed for (**A**). JSRV versus ENTV: *p* = 0.0002; JSRV versus Chm2-Egag: *p* = 0.4031 (n/s); JSRV versus Chm3-Eenv: *p* = 0.0001; JSRV versus Chm4-EU3: *p* = 0.0009. * *p* < 0.001.

**Figure 4 viruses-11-01061-f004:**
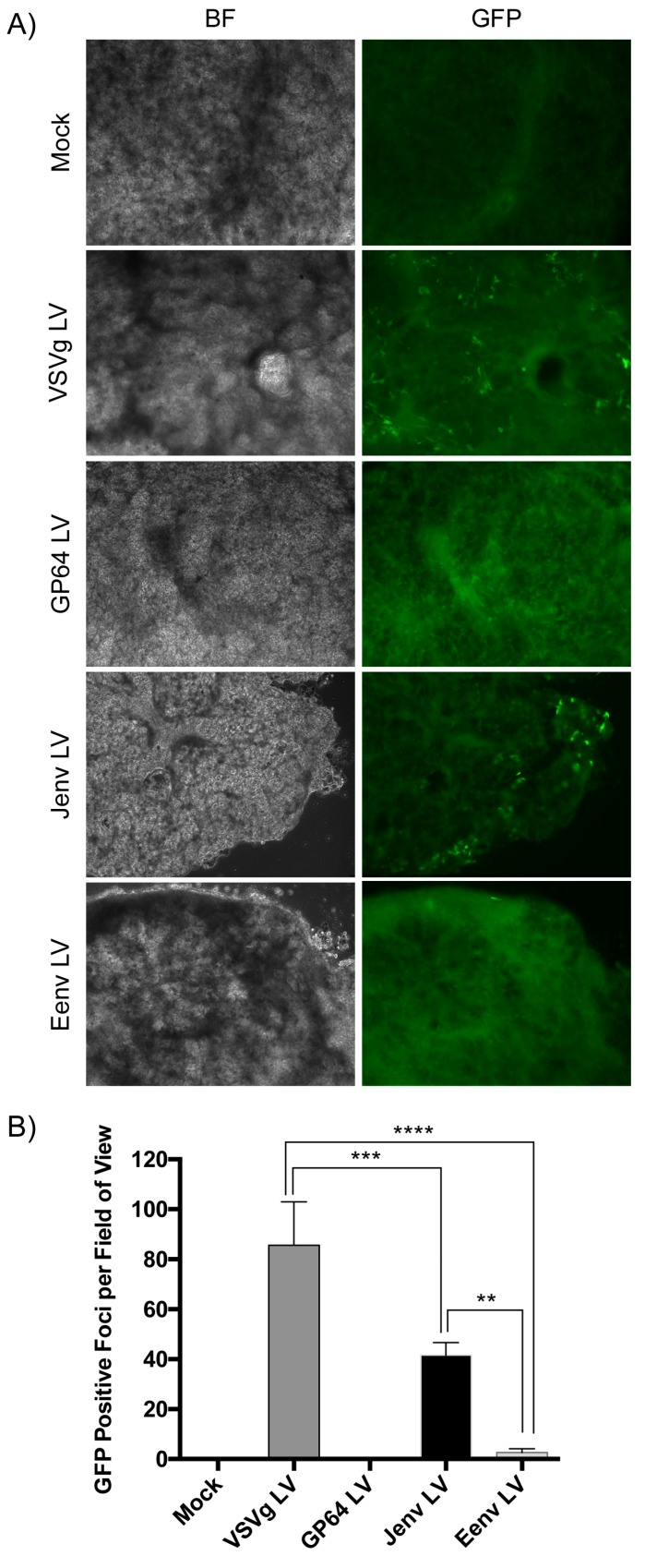
Jenv mediates lentivirus vector (LV) entry to ovine lung tissue, whereas Eenv does not. (**A**) Representative images of ovine lung tissue slices transduced with 1 × 10^6^ IFU of GFP-expressing LVs pseudotyped with VSVg, GP64, Jenv, and Eenv envelope glycoproteins. Forty-eight hours post-transduction, the lung tissue slices were sandwiched between a coverslip and glass slide and imaged with an inverted fluorescence microscope. Jenv and VSVg-pseudotyped LV transduction was visualized by the punctate green foci in the GFP channel. (**B**) Three fields of view imaged at 400× magnification were quantified per vector. Quantification of the total number of GFP-positive cells per field of view revealed significant differences between VSVg and all other LVs (**** *p* < 0.0001), as well as between Jenv and all other LVs (*** *p* < 0.001, with the exception of Eenv LV; ** *p* < 0.01) using one-way ANOVA.

**Figure 5 viruses-11-01061-f005:**
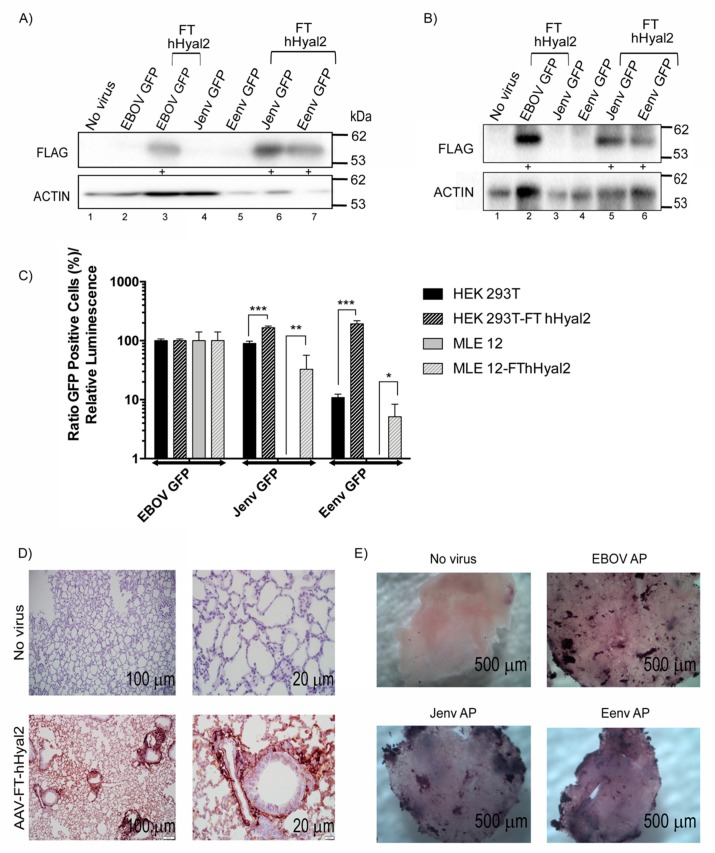
Overexpression of Hyal2 greatly enhances Eenv-mediated cell entry. (**A**) Western blot analysis of lysates from HEK 293T cells transfected with FLAG-tagged human Hyal2 (FT hHyal2 transfected lanes denoted with +) and transduced with 1 × 10^6^ IFU GFP-expressing lentivectors pseudotyped with EBOV, Jenv, and Eenv. (**B**) Western blot analysis of lysates from MLE 12 cells transfected with FLAG-tagged human Hyal2 (FT hHyal2 transfected lanes denoted with +) and transduced with 1 × 10^6^ IFU GFP-expressing lentivectors pseudotyped with EBOV, Jenv, and Eenv. (**C**) Data showing the ratio between GFP flow cytometry values and luminescence values from HEK 293T and MLE 12 cells co-transfected with FT hHyal2 and a plasmid expressing the firefly luciferase reporter gene, and then transduced with the LVs described in (**A**) 48 h post-transfection. Values were normalized to those obtained with EBOV LV and analyzed by two-way ANOVA (* *p* < 0.01, ** *p* < 0.001, *** *p* < 0.0001). (**D**) Immunohistochemical staining of murine lung tissue slices from mice transduced with 1 × 10^11^ vg of AAV6.2FF-FT-hHyal2 with a rabbit anti-FLAG antibody. (**E**) Lung slices from AAV6.2FF-FT-hHyal2 infected mice transduced with 1 × 10^6^ IFU of human placental alkaline phosphatase (hPLAP) expressing LVs pseudotyped with EBOV, Jenv, and Eenv and stained 48 h later.

**Figure 6 viruses-11-01061-f006:**
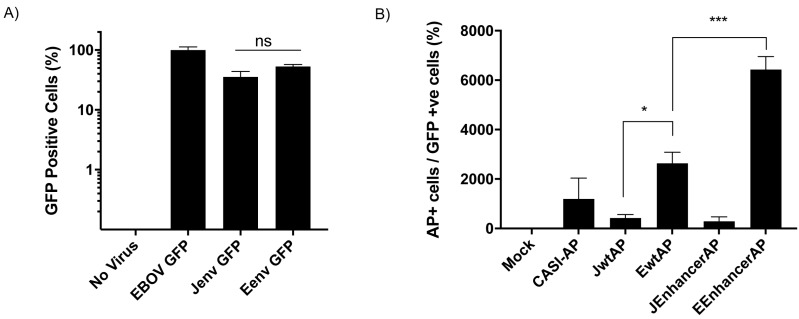
The ENTV LTR is significantly more active than the JSRV LTR in primary ovine chondrocytes. (**A**) Flow cytometry data showing the percentages of ovine chondrocytes expressing GFP 48 h after transduction with 1 × 10^6^ IFU of EBOV, Jenv, or Eenv pseudotyped LVs. Values were normalized to those obtained with EBOV LV. A two-way ANOVA was performed for Jenv and Eenv samples of ovine primary chondrocyte cells (not significant *p* = 0.5622). (**B**) Ratio of the count of AP-positive foci and the percentages of GFP-expressing cells in primary ovine chondrocytes co-transfected with pCASI-GFP plus AP expressing plasmids under the control of the JSRV LTR (JwtAP), the ENTV LTR (EwtAP), the JSRV LTR + JE enhancer (JEnhancerAP), or the ENTV LTR + EE Enhancer (EEnhancerAP). A Student’s unpaired, two-tailed *t*-test was performed between JwtAP and EwtAP (* *p* = 0.0124) and between EwtAP and EenhancerAP (*** *p* = 0.0034).

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
