# Peer review of "The U3 and Env Proteins of Jaagsiekte Sheep Retrovirus and Enzootic Nasal Tumor Virus Both Contribute to Tissue Tropism"

_viruses, 2019, doi:10.3390/v11111061_

Round 1

Reviewer 1 Report

The manuscript was correctly revised, but has still several problems. In the previous review, I just mentioned critical comments. In the current review, I pointed detailed issues to make the manuscript more understandable for me.

The authors answered that the chm6 virus cannot replicate to my previous comment. If so, the chm6 virus cannot be compared with other chimeras. Furthermore, viral protein was not detected in lung tissue slices inoculated with the chm6 virus (Figure 2). The chm6 virus was not used in other experiments. The chm6 virus is almost same as the chm3 virus. Thus, I recommend that the chm6 virus is removed from the manuscript. In Figure 2, only one image in the each chimera virus is indicated. Quantitative analysis or additional images are required to conclude so. In Figure 3A, are there significant differences between the chimera viruses in RT activity? “Given the results of the ex vivo ovine nasal turbinate tissue slice experiment, we were interested to determine whether overexpression of Hyal2 could enhance Eenv-LV entry (line 368-369).” I do not understand the reason. In Figure 5D and E, empty AAV infection is the best control, although the authors show in this manuscript that lung tissue is less susceptible to Eenv LV infection than Jenv LV infection using GFP reporter (Figure 4). In the previous review, I commented that the authors should analyze endogenous Hyal2 expression levels in lung and nasal cells. The authors answered that there is no good antibody against Hyal2. I would like to know mRNA levels of Hyal2 in those cells.

Author Response

Reviwer#1:
The manuscript was correctly revised, but has still several problems. In the previous review, I just mentioned critical comments. In the current review, I pointed detailed issues to make the manuscript more understandable for me.

1. The authors answered that the chm6 virus cannot replicate to my previous comment. If so, the chm6 virus cannot be compared with other chimeras. Furthermore, viral protein was not detected in lung tissue slices inoculated with the chm6 virus (Figure 2). The chm6 virus was not used in other experiments. The chm6 virus is almost same as the chm3 virus. Thus, I recommend that the chm6 virus is removed from the manuscript.

We agree that removing chm6 may add some clarity to the manuscript so it has been removed. Thank you for the suggestion.

2. In Figure 2, only one image in the each chimera virus is indicated. Quantitative analysis or additional images are required to conclude.

We have revised the figure legend to indicate that these are “representative” images. Immunohistochemical staining is not easily quantifiable thus we have included this figure more for descriptive purposes. Qualitative data is provided in subsequent figures when evaluating pseudotyped viruses.

3. In Figure 3A, are there significant differences between the chimera viruses in RT activity?

Yes, there are significant differences. These data are included in the figure legend.

4. “Given the results of the ex vivo ovine nasal turbinate tissue slice experiment, we were interested to determine whether overexpression of Hyal2 could enhance Eenv-LV entry (line 368-369).” I do not understand the reason.

Thank you for catching this. It should say, “Given the results of the ex vivo ovine lung slice experiment …”. This has been corrected.

5. In Figure 5D and E, empty AAV infection is the best control, although the authors show in this manuscript that lung tissue is less susceptible to Eenv LV infection than Jenv LV infection using GFP reporter (Figure 4).

AAV expressing an irrelevant reporter gene such as GFP would have been a better control; however, PBS controls are used in many reports as AAV vectors are expensive and laborious to produce so there is precedent in the literature.

6. In the previous review, I commented that the authors should analyze endogenous Hyal2 expression levels in lung and nasal cells. The authors answered that there is no good antibody against Hyal2. I would like to know mRNA levels of Hyal2 in those cells.

This would be an informative experiment and should we have access to more fetal ovine tissues, we will harvest samples for RNA analysis.

Reviewer 2 Report

This revised manuscript describes the characterization of molecular chimeras between Jaagsiekte sheep retrovirus (JSRV) and enzootic nasal tumor virus (ENTV).  The authors have attempted to map regions of these betaretroviral genomes that contribute to tissue specificity.  The manuscript has been improved, particularly some of the quantitative analysis, and the results are interesting.  However, several other important issues need to be addressed to clarify interpretation of the results.

Major comments:

It would be helpful to put the enhancer element on the map in Fig. 1A. 5A – The levels of actin are quite variable in this Western blot, and some explanation should be given. 5A and B – The marking of these figures is quite confusing. Use headings to show where FLAG-tagged hHyal2 is expressed and which lanes have been transduced with a particular vector. 5 C – Again the figure is confusing. Use a line underneath the graph to indicate which Env-GFP is being expressed. Section 3.6 – Most retroviruses have enhancers within their U3 regions. However, if alternative enhancers are being analyzed, the readers need to be informed about their nature and location before results of chimeras can be interpreted.  JSRV has a Rev-like Rej protein. Does ENTV have such a protein and does this affect the chimera infection results? The authors have attempted to address the previous criticism that the tissue specificity of other retroviruses was not discussed. The revised Discussion is quite superficial and rambling.  Description of results from the most closely related retrovirus (a betaretrovirus, MMTV) cites a single reference from 1993. The Discussion needs to be more concise and related back to the results described here.

Minor comments:

Line 104 – Define “vg”. Line 116 – “preplaced” should be “replaced”. Line 281 – “ELSIA” Line 379 – “GPF” Line 416 – “that” should be “than”. Line 560 – “reagion” At least two different fonts are used in the manuscript.

Author Response

Reviewer #2:
This revised manuscript describes the characterization of molecular chimeras between Jaagsiekte sheep retrovirus (JSRV) and enzootic nasal tumor virus (ENTV). The authors have attempted to map regions of these betaretroviral genomes that contribute to tissue specificity. The manuscript has been improved, particularly some of the quantitative analysis, and the results are interesting. However, several other important issues need to be addressed to clarify interpretation of the results.

Major comments:
1. It would be helpful to put the enhancer element on the map in Fig. 1A.

We have included an additional figure as part of Figure 1 which defines the boundaries of the JE/EE enhancer region.

2. 5A – The levels of actin are quite variable in this Western blot, and some explanation should be given.

The purpose of Figure 5A is to show that lysates from HEK 293T cells transfected with FT hHyal2 and transduced with EBOV, Jenv and Eenv pseudotyped lentivectors did in fact express FT Hyal2. Since we normalized for transfection efficiency using a luciferase expressing plasmid, the lanes in the WB were loaded with equal volumes of cell lysate rather than total protein.

3. 5A and B – The marking of these figures is quite confusing. Use headings to show where FLAG-tagged hHyal2 is expressed and which lanes have been transduced with a particular vector. 5C – Again the figure is confusing. Use a line underneath the graph to indicate which Env-GFP is being expressed.

Agreed. We have modified the way in which the lanes are labelled.

4. Section 3.6 – Most retroviruses have enhancers within their U3 regions. However, if alternative enhancers are being analyzed, the readers need to be informed about their nature and location before results of chimeras can be interpreted.

Good point. We have provided some background information to educate the readers about
the nature of the JSRV/ENTV enhancer sequence.

5. JSRV has a Rev-like protein. Does ENTV have such a protein and does this affect the chimera infection results?

The JSRV Env signal peptide (SP) has been shown to bind to the signal peptide response element (SPRE) located at the 3’ end of env and UTR region and facilitate unspliced RNA export. The nuclear export signal, nuclear localization signal (NLS), and arginine rich motif of the JSRV signal peptide are involved the RNA export function of the JSRV SP. Unlike JSRV, ENTV Env does not appear to encode a NLS within its SP domain nor are there any reports in the literature suggesting that the SP of ENTV Env functions similarly to that of JSRV Env. Although deletion and/or mutation of the JSRV Env SP or SPRE reduces virus
production compared to wildtype, we always infected tissues/cells with the same number of infectious units for each of the chimeras.

6. The authors have attempted to address the previous criticism that the tissue specificity of other retroviruses was not discussed. The revised Discussion is quite superficial and rambling. Description of results from the most closely related retrovirus (a betaretrovirus, MMTV) cites a single reference from 1993. The Discussion needs to be more concise and related back to the results described here.

We have revised this section of the discussion to make it more concise.

Minor comments:
Line 104 – Define “vg”.
Line 116 – “preplaced” should be “replaced”.
Line 281 – “ELSIA”
Line 379 – “GPF”
Line 416 – “that” should be “than”.
Line 560 – “reagion”
At least two different fonts are used in the manuscript.

Thank you for pointing out these errors. They have been corrected.

This manuscript is a resubmission of an earlier submission. The following is a list of the peer review reports and author responses from that submission.

Round 1

Reviewer 1 Report

Two highly related betaretroviruses Jaagsiekte sheep retrovirus (JSRV) and enzootic nasal tumor virus (ENTV) cause tumors in different parts of the respiratory tract of small ruminants. This manuscript addresses the regions of the ENTV genome that are responsible for disease specificity using molecular chimeras. Starting with the infectious JSRV clone, substitutions with various regions of the ENTV genome have been performed. Viruses produced from these clones then were used for infections of lung or nasal tissue slices. Reporter lentivectors also have been used for pseudotyping with different viral envelope proteins followed by infections of 293T or either lamb testes or murine lung cells. The authors conclude that both the envelope and LTR regions are important for disease specificity, yet they do not provide background or comparisons to similar studies from other retroviruses. Nevertheless, these experiments appear to be carefully performed. Nasal turbinate chondrocytes also are identified as an ENTV cell target. Most importantly, they have used physiological conditions with live tissues to address this very difficult and interesting viral system.

Several points would improve this paper.

Major points:

The authors do not compare their work to other more characterized retroviruses in the Discussion.  Studies with other retroviruses have shown that LTR-driven transcription and the viral envelope affect viral tropism. The Introduction/Discussion should mention the characterization of the ENTV enhancer. Is this actually within envelope or where the env gene and the U3 region of the LTR overlap? Statistical analysis of most data has been performed. However, the lung and nasal turbinate slice infections were not statistically analyzed (Figs. 5 and 6). Statistical analysis for Fig. 7B is not indicated on the figure. Fig.1 – Very little p27 (CA?) is expressed for the chimeras 4, 5, and 6, and very low levels of Env (SU?) are made for chimeras 3 and 6.  Because the gel is extensively cropped, the levels of precursors cannot be determined. How does this correlate with infectivity?  Or is this cell-type dependent? Have the packaging sequences for JSRV and ENTV been defined?

Minor points:

Line 120 – “preplaced” Table 1 could be moved to supplementary data. Number the lanes on Western blots. Fig. 2B – Lane loading is not equal, especially for EBOV-GFP.  Perhaps the gels could be scanned and the relative levels determined.  Are the JSRV and ENTV envelope proteins cytotoxic? Line 340 – Reword this sentence.  The current version implies that ENTV Env exceeds the efficiency of transduction by other Env proteins. Fig. 3 legend – There is no need to repeat p values in the legend if they are in the figure. What are the promoters for the control vectors? What is CASI-AP? Are standard deviations indicated by error bars? How many experiments were performed? The legend for Fig. 7 Is confusing, particularly parts D-F, and should be rewritten. (D) appears twice. Fig. 7F would be more clear by showing the constructs together with the results.

Reviewer 2 Report

Rosales Gerpe et al. found that Env proteins and LTRs of JSRV and ENTV are responsible for their different tissue tropisms. Although this result is interesting, the manuscript is not understandable for me and has many problems. Major comments are described as follows.

(1) Protein bands of Env and Gag p27 were faint in the chm6 virus (Figure 1B). The chm6 cannot proliferate?

(2) Level of Env SU protein in the chm3 virus was lower than those of the chm1, 2, and 4, but levels of Gag p24 in these chimera viruses were similar (Figure 1B). Why? Eenv protein is less efficiently incorporated into viral particles than Jenv? Thus, Eenv-containing viruses require higher amount of the receptor protein?

(3) GFP expression levels in different vectors are different (Figure 2B). The GFP-encoding LV vector genome contains different promoters? Please explain the reason.

(4) GFP expression level in Jenv LV was higher than that in Eenv LV (Figure 2B lower panel). It suggests that virion number of the Jenv vector was higher than that of Eenv vector. Thus, transduction titers of these vectors cannot be compared in Figures 3 and 4. These vectors should be normalized as in Figures 5. In Figure 7, transduction titers were normalized by transfection efficiency, but it should be normalized by RT.

(5) Promoter activity of JSRV and ENTV was analyzed in primary ovine chondrocyte cells and it was found that ENTV promoter was more active than JSRV promoter in chondrocytes (Figure 7F). It should be additionally performed in primary lung cells to assess whether JSRV promoter is more active than ENTV promoter in lung cells.

(6) Quality of GFP images is poor (Figures 3B and 4). Green signal is detected in GP64 and Eenv vector-transduced tissues (Figure 4). In merge images of Figures 3B and 4, I can see strong spread green signal.

(7) It was concluded that high levels of Hyal2 are crucial for infection by Eenv. The authors should analyze endogenous Hyal2 expression levels in lung and nasal cells in this study.

(8) Jenv vector efficiently infected HEK293T cells without FThHyal2 (Figure 7). It suggests that Jenv can infect both of lung and nasal cells. However, JSRV does not induce nasal cancer, because JSRV promoter is not active in nasal cells. If so, the chm4 and 5 viruses that contain JSRV Env and ENTV promoter should replicate in nasal cells. However, only the chm2, 3, and 4 viruses were analyzed in Figure 6. The chm1, 5, and 6 viruses should be analyzed.

(9) From the result of Figure 7, HEK293T cells were more susceptible to Jenv-mediated infection than Eenv-mediated infection, suggesting that JSRV more efficiently replicate in HEK293T cells than ENTV. However, Eenv level was higher than Jenv level (Figure 1B left panel). These results are inconsistent.

(10) In Figure 3C, MLE12-FThHyal2 cells were analyzed. Mock-transfected MLE12 cells should be used as control.

(11) Eenv LV infected FThHyal2-transfected HEK293T cells similar to Jenv LV (Figure 7), but Eenv LV less efficiently infected FThHyal2-transfected MLE12 cells than Jenv LV (Figure 3). These results are inconsistent.

(12) Lung tissue slices were inoculated with LV and chimera viruses (Figures 4 and 5). Nasal turbinate slices were inoculated with chimera viruses (Figure 6) but not LV. LV should be also inoculated to nasal turbinate slices to confirm whether Jenv LV can infect nasal cells.